# Patients’ Desire for Psychological Support When Receiving a Cancer Diagnostic

**DOI:** 10.3390/ijerph192114474

**Published:** 2022-11-04

**Authors:** Tomás Blasco, Esther Jovell, Rosanna Mirapeix, Concha Leon

**Affiliations:** 1GIES, Grup d’Investigació en Estrès i Salut, Departament de Psicologia Bàsica, Universitat Autònoma de Barcelona, 08193 Barcelona, Spain; 2Hospital de Terrassa, Unidad de Epidemiología y Evaluación Asistencial, 08227 Terrassa, Spain

**Keywords:** cancer patients, psychological support, coping, mood disorders

## Abstract

Background: Factors related to the desire of receiving psychological help in cancer patients are not well known. The aim of this study is to assess the prevalence of patients who would ask for psychological assistance in the first weeks following diagnosis, and to identify their psychosocial and disease-related profile. Method: This cross-sectional study assessed 229 consecutive cancer outpatients at a visit with their oncologist to be informed about the treatment they will receive. Disease-related and medical characteristics were assessed, and patients were asked about their mood states, levels of self-efficacy, and difficulties coping with the disease. Finally, patients were asked about their desire to receive psychological assistance. Results: Only 20% of patients expressed a desire for psychological help. These patients were lower in age and had previous history of mood disorders and reported higher discouragement and coping difficulties. These variables explained 30.6% of variance. Conclusions: Although psycho-oncologists can provide helpful interventions, the percentage of patients interested in receiving psychological assistance in this study is low. Although further studies are needed, results from this study suggest methods that could easily be used by oncologists and nurses to identify patients who would like to receive psychological support.

## 1. Introduction

Health problems can be produced by biological (i.e., cancer, diabetes) or psychological (emotional distress) reasons and can evolve as an acute or a chronic process. Whatever the case, many non-pharmacological interventions (NPIs) can help patients to improve their quality of life if these problems appear. Furthermore, NPIs can be used not only to treat health problems once they appear, but also to prevent them. Thus, NPIs’ efficacy can be enhanced if they are applied both to reduce risk factors before the illness appears and to provide help at the first stages of a disease, when a diagnosis is given to the patient. When a person knows that they have a serious disease with an uncertain prognosis, the patient’s life is threatened and emotional distress appears. Psychological interventions are useful NPIs to enable patients to cope with this emotional distress and to prevent psychopathology. Thus, psychological support should be provided early to patients suffering serious illnesses. Cancer belongs to this group of diseases; it is diagnosed in millions of people each year, and it is estimated that its incidence will increase. Public health policies should take into account this situation and should plan interventions providing psychological support to patients in order to reduce emotional distress and, hence, years lived with a disability. This psychological support should be provided as soon as the disease has been diagnosed, since a cancer diagnosis produces emotional distress in patients and their families [1,2,3,4,5]. Depending on the study, the incidence of clinically significant distress could range between 24% [6], 52% [7] and 61% [8]. These differences may be explained by variations in the specific type of cancer and/or whether patients were assessed at diagnosis or while receiving oncological treatment. Therefore, previous evidence suggests that many cancer patients might need assistance to cope with their emotional distress.

Cancer patients could benefit from psycho-oncological support, which is useful in reducing anxiety, depression, and distress in cancer patients [9,10,11]. However, less than 30% of patients with high levels of emotional distress ask for psycho-oncology assistance [5,12,13,14,15,16]. Some reasons have been proposed to understand the low demand for support. These include that some patients may not need psychological assistance because their levels of distress, depression, and anxiety may be transient and will decrease over time [17], whereas other patients may think they are able to cope with the cancer experience using their own strategies and psychological resources [18,19,20,21].

In addition, some studies have found a relationship between levels of distress and desire for help [22,23], but this correlation has not been observed in other studies [13,20,24,25,26]. Thus, there are reasons other than distress levels that are considered for psycho-oncological assistance [27].

In this sense, younger patients seem more likely to accept psychological support, Refs. [25,26,27,28,29] and more women than men demand psychological assistance [25,30,31]. Nevertheless, cancer type does not relate to the demand of psychological support [25].

Finally, it has been found that the type of psychological support offered is also related to the desire for psychological help. Studies that offered patients a chance to engage in psychosocial support groups had rates of participation between 21% and 66% [20,26,29,30]. However, when asked if they wanted to receive psychological support without explaining what kind of support it would be, only 20% of patients were engaged [24,25]. Furthermore, it seems that while online and group therapy are acceptable for patients, individual face-to-face therapy is preferred [32].

Thus, further research is needed to know which patients would ask for psychological assistance as well as the prevalence of these patients. This knowledge might help to optimize psycho-oncological services, since it would provide physicians and nurses with a guide to identify patients who need psychological support.

In this context, the aim of this study is to assess the prevalence of patients who would ask for psychological assistance after a cancer diagnosis, and to identify their psychosocial and disease-related profile.

## 2. Materials and Methods

### 2.1. Patients and Setting

This study was developed at the medical-oncology department at the Cancer Center of Hospital de Terrassa and was approved by its Ethics Committee. Outpatients who met the inclusion criteria (patients had to be at least 18 years old and free of any cognitive impairment) and were receiving their second visit with their oncologist, to be informed about their oncological treatment, were included in the study. These patients were aware of their diagnoses from a previous visit no more than two-three weeks prior. During this second visit, patients were informed about the study and its voluntary nature, that the data would only be used for research purposes, and that all information collected was anonymous. They did not start completing it without having first given explicit consent. A sample of 355 patients was recruited.

### 2.2. Instruments and Procedure

The oncologists completed a separate sheet registering sociodemographic (gender, age) and clinical data for each patient (kind and stage of tumor and presence of mood disorders across the life span).

The oncologist asked the patients six questions, and they answered them using a categorial scale of four points ("not at all,” “not so much,” “much,” “very much”). Four questions were addressed to assess how the patient’s mood state was during the last week (asking about levels of discouragement, nervousness, fear, and anger). The other two questions asked about: (a) the actual levels of self-efficacy to cope with the situation, and (b) difficulties to cope with the situation.

In order to develop a logistic regression analysis (see Statistical Procedure below), answers were recorded in two categories: “YES” (when patients stated “much” or “very much”) and “NO” (when patients stated “not at all” or “not so much”). Thus, it was considered that a patient has discouragement (YES) if they answered “much” or “very much.” If they answered “not at all” or “not so much,” it was considered that the patient did not have discouragement (NO). The same procedure was applied for nervousness, fear, anger, self-efficacy, and coping difficulties. Finally, the oncologist asked the patient if they would want to receive psychological assistance. Patients answered this question by saying “yes,” “no,” or “I do not know.”

### 2.3. Data Analysis

On the final sample of 229 patients (see Participants section), a univariate analysis of the relationship between patients’ desire for psychological help, education, living alone, cancer site, and presence of metastatic disease was developed using a chi-squared test. Neither education (*p* = 0.663), living alone (*p* = 0.100), cancer site (*p* = 0.155), nor presence of metastatic disease (*p* = 0.141) had a relationship with the desire to receive psychological help.

Thus, a regression logistic procedure (Forward Wald Method) was developed to assess the relationship between age, gender, mood states (discouragement, nervousness, fear, anger), self-efficacy, coping difficulties, and previous history of mood disorders. All variables with a univariate *p*-value less than 0.05 (*t*-test for age, and chi-squared test for gender, mood states, self-efficacy, coping difficulties, and previous history of mood disorders) were entered in the regression model.

## 3. Results

### 3.1. Participants

Table 1 shows the psychosocial profile of the 355 patients recruited. For the “Yes” subsample, mean age was 57.41 (SD = 13.32), for the “No” subsample, mean age was 62.30 (SD = 13.92) and for the “I do not know” subsample, mean age was 60.38 (SD = 12.38). Only patients who answered “Yes” or “No” to the question: “Do you want to receive psychological help?” were included in the logistic regression analysis. Thus, the 126 patients (35.4% of the total sample) who answered “I do not know” were excluded. Finally, data from 229 patients (115 women and 114 men) were used for the analysis.

### 3.2. Patients Psychosocial Characteristics and Desire for Psychological Help

Table 2 shows the psychosocial profile of the subsample who answered “Yes” or “No” (n = 229). In total, 72 patients asked for psychological help (31.4% of the subsample), but if the whole sample of 355 patients is considered, this percentage is reduced to 20.2%.

### 3.3. Variables Associated with Desire for Psychological Help

Logistic Regression Analyses (see Table 3) showed that a desire for psychological help was associated with lower age (OR = 0.962), higher discouragement (OR = 0.330), higher coping difficulties (OR = 0.239), and previous history of mood disorders (OR = 3.013). The proportion of variance explained is 30.6%.

## 4. Discussion

The present study shows that an important number of patients (35.4% of the sample) do not know if they want to receive psychological support. These patients do not offer a clear different profile from patients who answered “yes” or “no.” Although this subsample included more breast cancer patients and less lung cancer patients than those observed in the other subsamples, in other features these patients were similar to those who answered “no,” such as mean age, level of education, and living alone. As discussed below, perhaps patients answered “I do not know” because the type of psychological support suggested was not specified.

On the other hand, the results of this research show that only 20% of patients of the whole sample of 355 participants were clearly interested in receiving psychological support. Previous studies have shown different prevalences to those found in our study, with higher rates which range from a 66% in a Norwegian breast cancer sample [20] to a 42.5% in a sample of melanoma patients, Ref. [26] and lower rates ranging 20–25% in studies with mixed or prostate cancer patients [14,24,25,29]. Furthermore, even lower rates (9–14%) were found by Dubruille et al. [28] in a sample of cancer patients aged more than 65 years.

Some arguments could explain these differences in rates of interest for receiving psychological support, as well as account for the relatively low rate observed in the present study. Firstly, it could be the type of psychological assistance offered. Perhaps some strategies (i.e., support groups vs. individual therapy) might be preferred by some patients, but not by others. In our study, when patients were asked if they wanted psychological support, the type of psychological help they would receive was not specified. This can lead to a confusion about what “psychological intervention” or “psychological support” means [33]. This may be of relevance here, since previous studies [24,32] have reported that the most common request for help was for face-to-face contact.

Secondly, it seems that lower rates of interest are observed when mixed cancer samples are used or when samples include only men, which is the case with prostate cancer patients. Some studies [28,30,31] have reported that men are less interested in psychological support since they believe they can take care of their problems independently [31,34]. However, in the present study, gender did not relate to a desire for psychological support. Although this relationship could exist, it may have been precluded by the associations between a desire for psychological help and the other variables found in the regression logistic model. In this sense, younger patients in our sample are more likely to engage in psychological assistance, and the OR found (0.96) is very similar to the range of OR (0.92–0.95) found in other studies [25,26,29,30]. Considering the present data, as well as those provided in previous research, it seems that age, gender, and type of tumor are related to the desire for receiving psychological assistance. However, age, gender, and type of tumor have a relationship between them. Thus, higher rates of interest for psychological support have been found in samples of breast cancer patients (a female cancer) [20], and lower rates in samples of prostate cancer patients (a male cancer) [29] or lung cancer (which has mainly male patients) [5]. If studies have assessed samples with mixed cancers, relationships between gender and type of tumor, and a desire for psychological support, it could preclude one to another. Furthermore, as age seems to be strongly related to a desire for psychological support, as it was found in the present and previous studies [25,26,29,30], it could also preclude the relationship between age, type of tumor, and a desire for psychological support. These considerations could help us to understand why the relationships between a desire for psychological support and gender, type of tumor, and age, are not always found across the studies.

In our sample, patients who would ask for psychological support were those with high levels of discouragement and coping difficulties, as well as those with previous history of mood disorders. Discouragement can be considered as a global measure of emotional distress that includes the other negative mood states assessed in this study (nervousness, fear, and anger). In this sense, emotional distress has also been related to a desire for receiving psychological support in previous studies using samples with mixed cancers and which did not specify the schedule of psychological treatment to be provided [14,23,24,25,30]. It is not surprising that patients who would ask for psychological support are those experiencing difficulties coping with cancer. More than this, these patients should also be those with lower self-efficacy, and this could be the reason why self-efficacy has not been included in the regression model. Finally, previous history of mood disorders also increases the desire for psychological help. This result can also be expected since these patients probably know that psychological assistance is useful [25].

Variance of a desire for psychological support explained by our model (30%) does not reach the 52% found in melanoma patients [26] but is higher than the 16% found by Dubruille et al. [28] in a sample of mixed cancer patients, such as ours. It seems that when only one type of tumor is considered, it is possible to more accurately identify patients who would accept psychological support. In this sense, our model is better than the model offered by Dubruille et al. [28], since it provides more suitable knowledge to identify cancer patients interested in receiving psychological support, in spite of their type and stage of tumor.

### Limitations

One of the limitations of this study could be that data were obtained using simple questions and not standardized procedures or questionnaires. However, we think that single questions provide valid information about patients’ feelings if one keeps in mind the time limitations that oncologists have in their visits, as well as the fact that some questionnaires require a previous knowledge to be correctly applied, which is hardly available to the physicians. On the other hand, single questions or Likert scales have been used in previous studies with cancer patients to assess the desire for psychological assistance or other psychological variables [28,35]. Finally, some other variables which have not been considered in the present study could also be related to levels of distress and hence increase the request of psychological help. Financial costs associated with diagnosis and care could increase the stressful disease experience and should be assessed in future studies, since it has been observed in cancer survivors that financial difficulties are related to poorer levels of mental health [36].

## 5. Conclusions

Further studies are needed to clarify when and why men and women differ in their preferences for scheduling psychological assistance, or which barriers perceived by patients and health professionals preclude the delivery of psychosocial care services [37,38]. There may still be a gap between the rate of patients who would benefit from psychological support and the rate of patients who ask for that or who are referred to psychosocial care by their oncologists. In the meantime, results of this research give a useful tool to physicians and nurses to identify patients who would be interested in receiving psycho-oncological support. A patient’s age and three simple questions (discouragement, coping difficulties, and history of mood disorders) provide the profile of those patients who would be interested in receiving psychological support. These patients can be quickly referred to the psycho-oncologist; thus, optimizing the cost-utility of healthcare services. However, future studies should be focused on developing useful strategies to provide care to patients who need psychological assistance but do not ask for this support.

## Figures and Tables

**Table 1 ijerph-19-14474-t001:** Psychosocial profile of participants who answered “Yes” (n = 72), “No” (n = 157) or “I do not know” (n = 126) to the question: “Do you want to receive psychological help?”

Variable	Yes	No	I Do not Know
	n (%)	n (%)	n (%)
Gender			
Female	43 (59.7)	72 (45.9)	74 (58.7)
Male	29 (40.3)	85 (54.1)	52 (41.3)
Education (1) (2)			
Primary school	44 (61.1)	100 (64.1)	83 (66.4)
Middle or superior level	28 (38.9)	56 (35.9)	42 (33.6)
Living alone (1)			
No	59 (81.9)	140 (89.7)	110 (87.3)
Yes	13 (18.1)	16 (10.3)	16 (12.7)
Cancer site (1)			
Breast	22 (30.6)	36 (23.1)	44 (34.9)
Lung	15 (20.8)	37 (23.7)	19 (15.1)
Colorectal	12 (16.7)	45 (28.8)	32 (25.4)
Others	23 (31.9)	38 (24.4)	31 (24.6)
Metastatic Disease (3)			
No	42 (66.7)	96 (69.6)	88 (73.9)
Yes	21 (33.3)	42 (30.4)	31 (26.1)

(1) There was 1 missing case in the “No” subsample. (2) There was 1 missing case in the “I do not know” subsample. (3) There were 9 missing cases in the “Yes” subsample, 19 missing cases in the “No” subsample, and 7 missing cases in the “I do not know” subsample.

**Table 2 ijerph-19-14474-t002:** Psychological characteristics of the subsample (n = 229).

Variables	Yes		No	
	n	%	n	%
Discouragement	70	30.6	159	69.4
Nervousness	103	45	126	55
Fear	46	20.1	183	79.9
Anger	55	24	174	76
Self-Efficacy	179	78.2	50	21.8
Coping Difficulties	85	37.1	144	62.9
History of Mood Disorders	47	20.5	182	79.5
Desire for Psychological Help	72	31.4	157	68.6

**Table 3 ijerph-19-14474-t003:** Characteristics of patients associated with a desire for psychological help: logistic regression model (n = 229). “Discouragement” and “Coping Difficulties” were codified as “high” if patients stated “much” or “very much,” and as “low” if patients stated “not at all” or “not too much.”

Variable	OR	95% CI Interval	*p*
Age	0.962	0.939 to 0.985	0.001
History of Mood Disorders	3.013	1.414 to 6.420	0.004
Discouragement	0.330	0.165 to 0.663	0.002
Coping Difficulties	0.239	0.121 to 0.474	0.000

## Data Availability

The data presented in this study are available on request for the corresponding author.

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
