# Peer review of "Patients’ Desire for Psychological Support When Receiving a Cancer Diagnostic"

_ijerph, 2022, doi:10.3390/ijerph192114474_

Round 1

Reviewer 1 Report

This paper is a well-written study and is considered useful in terms of providing the psychosocial disease-related profile of cancer patients who are motivated by psychological support to quickly refer them to psychological help.

I have some recommendations for revision.

1. Please provide the psycho-social profile of those who answered “I don’t know” or “not” to the question of whether they want to receive psychological assistance. Their profile is just as important as the profile of patients who would like to receive psychological help. In addition, the comparison of the psycho-social profile of the group that wants psychological help and the group that does not should be supplemented. These data would be useful in future research on strategies to encourage psychological help to the patients who would benefit from psychological support.

2. I recommended suggesting future studies that design the strategies for the patients in need of psychological support to benefit from psychological support in the conclusion paragraph.

3. In lines 186 to 190, the author stated that the type of cancer was not related to the desire for psychological assistance, it considered that the references and explanation for the above description are not clear. Please correct and supplement.

Author Response

Thank you very much for your valuous comments. We have introduced the following modifications in the paper.

1.A new Table has been added (now Table 1) showing the psychosocial profile of the three subsamples considered for the study.

This question has received some comments which have been placed in a  new paragraph, just at the beginning of the discussion section:

The present study showed that an important number of patients (35.4% of the sample) does not know if she/he wants to receive psychological support. These patients do not offer a clear different profile from patients who answered “yes” or “not”. Although this subsample included more breast cancer patients and less lung cancer patients than those observed in the other subsamples, in other features these patients were similar to those who answered “no” such as mean age, level of education and living alone. As it is discussed below, perhaps patients answered “I don’t know” because the type of psychological support suggested was not specified.

2. We have added the following statement just at the end of the Conclusion paragraph:

, but future studies should be focused in developing useful strategies to provide care to patients who need psychological assistance and do not ask for this support.

3. We have modified the last senteces of the third paragraph of the Discussion section (lines 205-209 in the first version of the paper; actually lines 249 to 262) as follows:

Considering the present data, as well as those provided in previous research, it seems that age, gender and type of tumor are related with desire for receiving psychological assistance. However, age, gender and type of tumor have a relationship between them. Thus, higher rates of interest for psychological support have been found in samples of breast cancer patients [20] (a female cancer), and lower rates in samples of prostate cancer patients [29] (a male cancer) or lung cancer [5] (which has mainly male patients). If studies have assessed samples with mixed cancers, relationships between gender and type of tumour and desire for psychological support could preclude one to another. Furthermore, as age seems to be strongly related with desire for psychological support as it was found in the present and previous studies [25, 26, 29, 30], it could also preclude the relationship between age, type of tumor and desire for psychological support.  These considerations could help us to understand why the relationships between desire for psychological support and gender, type of tumor, and age, are not always found across the studies.

Reviewer 2 Report

The authors mentioned in their Abstract Conclusion states "Conclusions: Although psycho-oncologists can provide helpful interventions, the percentage of patients interested in receiving psychological assistance in an Oncology Unit is low"
This statement can be read to mean that the scenario is similar in ALL Oncology Units. The authors should qualify this statement as "...In this Study..."

Author Response

Thank you for your comments. We have modified the statement as follows:

Although psycho-oncologists can provide helpful interventions, the percentage of patients interested in receiving psychological assistance in this study is low

Reviewer 3 Report

This study examined patient-level factors associated their desire of receiving psychological help among cancer survivors. While the authors did a good job of presenting the study objectives and results, there are a few concerns that need to be addressed. My specific comments are as follows: 

1. Does the study sample comprise of patients with a history of cancer? If that information is available, that maybe an important predictor of patients wanting to receive psychological assistance currently.

2. Both in the abstract and first part of the results section, it would be good if the authors could describe the study sample in one or two sentences.

3. It would be helpful if the authors presented the demographic (age, gender) and cancer-related information (cancer type, cancer stage) of the individuals in the study sample in Table 1.

4. Please report the univariate association of all demographic, cancer-related, and psychosocial characteristics in a table. That will help the readers better understand variable selection for the multivariable logistic regression model.

5. From table 2, it is not very clear what the reference category for discouragement or coping difficulties is. While this maybe explained in the text, tables should be stand-alone for ease of understanding for the readers. I would request the authors to edit the table to add more details - either in the table or in foot notes. 

6. One important point to consider here is the impact of financial hardship/toxicity associated with cancer diagnosis and care on patient's perceived need for psychological assistance. Cancer survivors who experience financial toxicity have greater odds of adverse mental health outcomes such as emotional problems. (see: Inguva et al. Financial toxicity and its impact on health outcomes and caregiver burden among adult cancer survivors in the USA. Future Oncol. 2022 Apr;18(13):1569-1581. doi: 10.2217/fon-2021-1282)

There are validated measures such as the COST measure that can be used to assess financial hardship/toxicity associated with cancer diagnosis and its treatment among cancer survivors (see: de Souza JA, Yap BJ, Wroblewski K, Blinder V, Araújo FS, Hlubocky FJ, Nicholas LH, O'Connor JM, Brockstein B, Ratain MJ, Daugherty CK, Cella D. Measuring financial toxicity as a clinically relevant patient-reported outcome: The validation of the COmprehensive Score for financial Toxicity (COST). Cancer. 2017 Feb 1;123(3):476-484. doi: 10.1002/cncr.30369)

While this study may not have collected information on patient's self-reported financial toxicity, this could be added as a limitation of the current study and as an implication for future research on identifying cancer survivors who would be interested in receiving psycho-oncological support.  

Author Response

Thank you very much for your valuous comments. We have introduced the following modifications in the paper.

  1. This is an interesting comment. Unfortunately, this information was not available for the present research. More than this, and unfortunately, 35 missing data were found concerning metastasic disease (see Table 1)
  2. We have modified the abstract to specify that the final sample had 229 cases. However, we have not modified the description of the whole sample, since we have included one additional table (actually Table 1) describing the psychosocial profiles of the patients who were not included in the logistic regression analysis (i.e. patients who answered “i don’t know to the question “do you want to receive psychological help?”)

  3. As it has previously commented, a new table has been added to the results section including this information as well as the information about the group that answered “I don’t know”.

  4. This information has been added in the Data Analysis section. The text has been modified adding a first paragraph as follows:

    On the final sample of 229 patients (see Participants section), a univariate analysis of the relationship between patients’ desire for psychological help, education, living alone, and cancer site and presence of metastasic desease was developed using chi-squared test. Neither cancer site (p=.155), education (p=.663), living alone (p=.100) nor presence of metastasic disease (p=.141) had relationship with desire for receiving psychological help.

  5. This table is now Table 3, and the title has been modified as follows:

    Table 3.  Characteristics of patients associated with desire for psychological help: logistic regression model (n = 229). “Discouragement” and “Coping difficulties” were codified as “high” if patients stated “much” or “very much” and as “low” if patients stated “not at all” or “not too much”.

  6. We have considered this question adding some sentences (as well as the reference provided at the end of the limitations section

    Finally, some other variables which have not been considered in the present study could also be related with levels of distress and hence increase the request of psychological help. Financial costs associated to diagnosis and care could increase the stressful disease experience and should be assessed in future studies, since it has been observed in cancer survivors that financial difficulties are related with poorer levels of mental health [36].

Round 2

Reviewer 1 Report

The requested recommendations have been addressed sufficiently. Thank you for your hard work.